# Brain Health and Cognition in Older Adults: Roadmap and Milestones towards the Implementation of Preventive Strategies

**DOI:** 10.3390/brainsci14010055

**Published:** 2024-01-06

**Authors:** Federico Emanuele Pozzi, Giulia Remoli, Lucio Tremolizzo, Ildebrando Appollonio, Carlo Ferrarese, Luca Cuffaro

**Affiliations:** 1School of Medicine and Surgery, University of Milano-Bicocca, 20100 Milan, Italy; g.remoli@campus.unimib.it (G.R.); lucio.tremolizzo@unimib.it (L.T.); ildebrando.appollonio@unimib.it (I.A.); carlo.ferrarese@unimib.it (C.F.); luca.cuffaro@unimib.it (L.C.); 2Neurology Department & Brain Health Service, Fondazione IRCCS San Gerardo dei Tintori, 20900 Monza, Italy; 3Milan Center for Neuroscience (Neuro-MI), University of Milano-Bicocca, 20126 Milan, Italy

**Keywords:** subjective cognitive decline, brain health, prevention, Alzheimer’s disease, cognition

## Abstract

In this narrative review, we delve into the evolving concept of brain health, as recognized by the WHO, focusing on its intersection with cognitive decline. We emphasize the imperative need for preventive strategies, particularly in older adults. We describe the target population that might benefit the most from risk-based approaches—namely, people with subjective cognitive decline. Additionally, we consider universal prevention in cognitively unimpaired middle-aged and older adults. Delving into multidomain personalized preventive strategies, we report on empirical evidence surrounding modifiable risk factors and interventions crucial in mitigating cognitive decline. Next, we highlight the emergence of brain health services (BHS). We explain their proposed role in risk assessment, risk communication, and tailored interventions to reduce the risk of dementia. Commenting on ongoing BHS pilot experiences, we present the inception and framework of our own BHS in Monza, Italy, outlining its operational structure and care pathways. We emphasize the need for global collaboration and intensified research efforts to address the intricate determinants of brain health and their potential impact on healthcare systems worldwide.

## 1. Introduction 

The WHO defines brain health as “the state of brain functioning across cognitive, sensory, social-emotional, behavioral and motor domains, allowing a person to realize their full potential over the life course, irrespective of the presence or absence of disorders” [1]. This is a dynamic definition, as it requires adaptations across the life course in all domains. Other definitions have been proposed, mostly in line with the one endorsed by the WHO [2]. As individuals age, cognition becomes a pivotal determinant of brain health, albeit not the sole factor. Therefore, preventing and addressing cognitive impairment is an imperative objective to supporting brain health, bearing in mind that this process should ideally start at birth and continue throughout the whole life. This should be achieved through a combined implementation of public health policies aimed at reducing factors detrimental to brain health in the general population (such as poverty, lack of education, and racial discrimination [3,4]) and a shift of healthcare services towards early detection of at-risk individuals and personalized prevention strategies [5]. Such new policies should be synergistically developed at a global level to avoid further inequality [6]. 

The current paper will focus on the cognitive aspects of brain health in older adults, reviewing the most significant advancements in the prevention of cognitive impairment and providing an overview of future steps in the field. We will finally report on the inception of our personal experience. 

## 2. Brain Health as a New Target 

The concept of brain health, defined as the “state of physical, mental, and social well-being”, was first articulated by the WHO in its constitution in the late 1940s. Despite its longstanding presence in health discourse, the lack of a universally accepted definition and the absence of standardized, objective quantitative methods have posed significant challenges, hindering its practical application in real-life settings. This need for a more cohesive understanding and measurement of Brain Health has been emphasized by Hachinski et al. [7].

Recent developments over the past two years have witnessed a notable shift in recognizing brain health as a paramount concern. Various influential associations, including the WHO, the American Heart Association [8], and the European Academy of Neurology [9], have acknowledged and prioritized brain health on their agendas. This growing recognition reflects an increasing understanding of the profound impact of brain well-being on overall health and quality of life.

Epidemiological evidence has further reinforced the importance of brain health, indicating that approximately 90% of all strokes are attributable to a few potentially modifiable risk factors. Therefore, primary prevention is vital to curb the high burden of stroke [10]. Similarly, about 40% of dementia cases can be attributed to modifiable lifestyle and cardiovascular risk factors [11]. The remaining cases are predominantly influenced by genetic factors, such as APOE ɛ4, and biological factors like the accumulation of amyloid and tau proteins. Additionally, other unknown risk factors and intricate interactions among these elements contribute to the complexity of understanding and preserving brain health.

During the Brain Health Summit organized by the European Academy of Neurology in Brussels in November 2023, the Brain Health Mission was launched to support the development of National Brain Health plans across Europe (https://www.ean.org/brain-health-mission, accessed on 28 November 2023). The burden of neurological diseases is the highest among non-communicable diseases and in economic society costs [12]. However, achieving brain health is a complex endeavor considering the limited number of breakthrough treatments for many neurological disorders and the limited scientific evidence in preventing neurological disorders [6].

Similarly, the 78th Session of the UN General Assembly (UNGA 78) in New York aimed to “rebuild trust and reignite global solidarity” by “accelerating action on the 2030 Agenda and its Sustainable Development Goals (SDG)”. As we approach 2030, emphasizing brain health will be crucial for countries to achieve the SDG of “ensuring healthy lives and promoting well-being for all ages” [13].

## 3. Preventive Strategies: Who and How 

Although preventive strategies should be ideally applied to the whole population, as there is no such thing as a null risk, resource constraints obligate the selection of targets that might benefit more from the implementation of preventive approaches. In this section, we will briefly discuss who should be the targets and the methods of these preventive strategies. 

### 3.1. Who: Subjective Cognitive Decline

According to the definition provided by an international group of experts almost a decade ago, subjective cognitive decline (SCD) is a condition characterized by a self-experienced persistent decline in cognition compared to the previously normal statuses, unrelated to acute events, with normal performances on demographically adjusted standardized cognitive tests used to classify mild cognitive impairment (MCI) [14]. Data from a large meta-analysis show that SCD is associated with a two-fold risk of developing MCI and dementia, but this risk is influenced by several factors, including recruitment source [15]. SCD may represent an early phase of AD cognitive changes (stage 2 of the NIA-AA research framework) [16], and a number of features are proposed that might increase the likelihood of SCD being due to preclinical AD. These features define the so-called SCD-plus and include, but are not limited to, subjective decline in memory, the onset of SCD within the last five years, the feeling of worse performance compared to peers, and the presence of the APOE ε4 genotype [14]. Another feature is the confirmation of the cognitive impairment, albeit minimal, by an informant; this may also give valuable insight into subtle changes in complex activities of daily living, which may not be captured by standard neuropsychological tests [17].

Data on SCD mainly come from large experiences in first-world countries, among which notable examples are given by the SCIENCe cohort in Amsterdam, the Netherlands [18] and the German DELCODE study [19]. These studies have enrolled hundreds of subjects and conducted impressive longitudinal follow-ups, sometimes spanning over a decade. However, one should bear in mind that SCD might have different prevalence and characteristics depending on the specific settings and socio-economics context, as shown by population studies in low-income countries [4]. 

SCD is a heterogeneous condition that has only recently been studied to find predictors of progression to MCI and dementia. These include also the use of CSF or PET biomarkers. In particular, SCD subjects with positive amyloid biomarkers show a steeper decline in cognition compared to patient with negative CSF biomarkers [20]. In this population a lower plasma Aβ42/Aβ40 ratio is associated with an increased risk of clinical conversion to MCI and brain amyloid deposition at 2-year follow-up [21]. At the same time, a positive brain amyloid-PET scan results in greater cognitive decline and medial temporal atrophy in the following 2 years [22]. These findings might justify the use of biomarkers even in a population otherwise considered “healthy”, as this provides important “short-term” prognostic information. However, it could be possible that in the near future, non-invasive and low-cost assessments might provide comparable predictive information with the aid of machine learning [23].

It may be difficult to differentiate between SCD and MCI, as this depends also on specific cut-offs used to diagnose the latter condition, which may result in different levels of “impairment” allowed in the condition of SCD. Although the current criteria for SCD imply normal performance on standardized neuropsychological tests, a considerable number of subjects may exhibit minor neuropsychological and/or behavioral deficits. These deficits could play a role in the progression to MCI. Moreover, it could be possible that new, more demanding tests might exhibit higher sensitivity for SCD [24]. However, depending on the cutoff, even standard neuropsychological tests may show interesting results. In the DELCODE cohort, 12% of SCD subjects exhibited minor neuropsychological deficits (as defined by a performance below 0.5 standard deviation on the total adjusted CERAD battery score). In these subjects, there was a four-fold-higher risk of progression to MCI compared to SCD without such deficits. In turn, SCD without minor deficits had a four-fold risk of progression to MCI compared to healthy controls without SCD. Interestingly, CSF AD biomarkers were significantly different in SCD with minor deficits compared to SCD without them (but the latter did not differ from healthy controls without SCD) [25]. It may be worth noticing that unimpaired controls were allowed to experience age-appropriate subjective subtle cognitive decline provided that they were not worried about it. This could be a controversial point, given that a subtle form of anosognosia or denial might be at play here, especially if such a decline could indeed be observed by an informant [14]. Indeed, the results from the INSIGHT-PreAD cohort show that low cognitive awareness in SCD may be a good marker of preclinical AD [26]. 

### 3.2. How: Multidomain Personalized Preventive Strategies 

According to the 2020 Lancet Commission report, up to 40% of dementia risk may be preventable by acting upon 12 modifiable risk factors [11]. These include diabetes, hypertension, traumatic brain injury, smoking, air pollution, midlife obesity, physical inactivity, depression, alcohol, hearing impairment, social isolation, and poor education. Poor sleep and oral health may be other risk factors, with increasing evidence accumulating in recent years [27,28,29]. The prevention potential and relative impact of these factors may vary based on the specific settings across different regions of the world [30]. Other factors are currently being investigated—such as substance abuse, microbiota and environmental features—although in certain cases, studies may suffer from biases related to reverse causality [31,32,33,34]. The relative contribution of each factor over the entire life course according to the 2020 Lancet Commission report is shown in Figure 1 even though we could not provide specific estimates for putative risk factors, such as poor sleep and poor oral health. 

Prevention of dementia may take the form of pharmacological and non-pharmacological strategies. Regarding the former, five trials with putative disease-modifying treatments are ongoing in preclinical populations [35], while comparatively more results have been reported for the latter. A few meta-analyses showed that non-pharmacological interventions in older adults with SCD seem to provide small, yet significant benefits on cognition, i.e., cognitive enhancement seems possible even when people appear cognitively normal [36,37]. In older adults (with or without SCD), multidomain personalized interventions may provide significant benefits in terms of both cognition and general health, as shown by the FINGER trial [38]; such interventions are also presumably cost-effective in preventing dementia [39], but further prospective data are warranted. However, other studies with multidimensional interventions have failed to show a clinical benefit, such as the MAPT and preDIVA trials [40,41]. Subsequent analyses of these studies have shown that the effect of multidimensional interventions was significant in subjects with CAIDE dementia risk scores of 6 or more and positive amyloid-PET in the MAPT study. In the preDIVA study, the incidence of dementia was almost half in the intervention group compared to the control group when the analysis was restricted to subjects with untreated hypertension who adhered to the interventions [42]. Regarding the FINGER trial, the significant effect already present in the whole sample was even more pronounced in APOE ε4 carriers, who tend to respond poorly to anti-amyloid treatments and develop side effects [43]. Moreover, the FINGER trial compared structured, high-intensity multidimensional interventions to generic health advice in the same domain and actually showed an improvement in both group, with a small, albeit significant, advantage for the intervention group [38]. All this evidence might in theory justify two different risk-reduction paradigms: on one hand, we could expect an improvement with generic health advice in the low-risk population, while subjects at high risk may benefit from structured multidimensional interventions. 

Multidomain interventions are mostly aimed at maintaining and improving cardiovascular fitness, including healthy diet, physical activity, and reduction of cardiovascular risk factors. A key mechanism through which such interventions could improve brain health and prevent neurodegenerative diseases is a positive effect on the neurovascular unit [5]. This is not surprising given the centrality of the neurovasculome to brain health and neurodegeneration [44]. The addition of sleep improvement could theoretically improve waste product clearance within the brain by enhancing the glymphatic system activity [29]. Further proposed mechanisms by which physical activity may improve brain health involve promotion of neurogenesis, decrease in neuroinflammation, increase in brain-derived neurotrophic factor, and increase in glymphatic flow [5]. 

On the other hand, brain health also involves the ability to adapt to and function despite increasing damage, a concept that partially overlaps with cognitive reserve [45]. While cognitive reserve may be primarily built through education and cognitive activities in the first part of life, higher education does not necessarily protect against late life cognitive decline [46]. On the other hand, brain resilience may be increased through cognitive training and socialization even in late adulthood [5]. Indeed, it has been shown that cognitive activities over the whole lifespan, including leisure activities in late life, may delay the onset of cognitive decline independent of early life education [47,48,49]. Moreover, factors linked to cognitive reserve, such as bilingualism and leisure-time cognitive activities in late life, are associated with cognitive function independently of brain volume [50]. In particular, it seems that bilingualism may act as a proxy of cognitive reserve, delaying dementia onset compared to monolingualism [51,52], although this finding was not widely replicated [46]. However, the potential confounding role of reverse causality, by which people with more education may choose to increase their cognitive activities, is notoriously difficult to address in clinical studies. Technological solutions such as mobile applications could be helpful for the implementation of cognitive training and educational strategies in the context of multidomain interventions, and may even outperform traditional paper-based approaches [53,54].

Aligned with the WHO Global Action Plan on the public health response to dementia for the period 2017–2025 [55], the World-Wide FINGERS (WW-FINGERS) Network was launched in 2017. The network aims to test the FINGER multidomain lifestyle model in various populations and settings [56]. With more than 25 countries joining the WW-FINGERS network, numerous projects are ongoing and several proposals are currently being evaluated [57].

In this context, we have recently launched the longitudinal Randomized Control Trial (RCT) Italian study on multidomain intervention in the at-risk elderly population, named In-TeMPO (Italian study with tailored multidomain interventions to prevent functional and cognitive decline in community-dwelling older adults). The study is set to last for two years and comprises two phases: an initial observational, population-based step, followed by recruitment for the longitudinal RCT efficacy phase. The data collection and core inclusion and exclusion criteria have been harmonized with the WW-FINGERS consortium, aiming for future joint analyses.

### 3.3. Beyond High-Risk: Universal Preventive Approach in Mid- and Latelife Cognitively Unimpaired 

While subjects with SCD present a higher risk of developing cognitive impairment and are therefore ideal targets of tailored preventive approaches, cognitively unimpaired middle-aged adults should not be ignored either. Indeed, observational evidence has showed that healthier lifestyle in late midlife is associated with overall better cognitive performance [58]. Multidomain intervention trials including cognitively unimpaired adults at midlife are ongoing [59]. A notable example of these is the Barcelona Brain Health Initiative, a prospective cohort studies enrolling hundreds of cognitively unimpaired middle-aged participants to evaluate and promote determinants of brain health [60]. In this cohort, the association of cardiorespiratory fitness and cardiovascular risk with cognition was mediated by cortical thickness, confirming the potential for preventive approaches also in middle aged subjects [61].

Since it might be impossible to implement personalized prevention in the whole population, brain health should be achieved through a universal preventive approach included in a public brain health agenda. This should include societal and political changes aiming at increasing physical activity, social integration, education and lifelong learning, cognitive activity, adopting a healthy diet, stopping smoking and reducing alcohol intake, and—with respect to elderly people—reducing the burden of chronic conditions and anticholinergic medications [62]. In middle-aged individuals, poor sleep may be an additional factor to target, as this is associated with lower brain health [63]. It is imperative that these changes are implemented at a global level, thus making brain health a worldwide priority with strategic and substantial investments [64]. Social and socioeconomic factors need also to be addressed to achieve an equitable approach to risk reduction since those are known to be associated with lifestyle risk factors [65,66]. 

Taken singularly, some of the aforementioned interventions also have the potential to modulate key biological aspects related to brain health. For instance, optimal intakes of vitamin B12, folates, omega-3 fatty acids, and antioxidants provided by adherence to the Mediterranean diet seem to reduce amyloid accumulation and decrease white matter hyperintensities [67,68]. Further benefits of healthy diet on cognition might be mediated by gut microbiota [69]. Physical activity could have neuroprotective effects through the modulation of myokines such as irisin/fibronectin type III domain-containing protein 5 in the hippocampus, which in turn increase levels of brain-derived neurotrophic factor [69]. Moreover, two systematic reviews of physical activity in middle and late life showed modest effects on structural and functional brain MRI measures [70,71]. Chronic alcohol abuse and nicotine increase neuroinflammation and oxidative stress, also affecting brain health through a detrimental effect on the microvascular bed [72,73]. In a study on the UK Brain Bank dataset, cognitive reserve proxies had significant moderating effects on structural brain integrity, acting as a buffering factor against neurodegeneration since midlife [74].

## 4. Brain Health Services and the Monza Experience 

Preventive strategies are now implemented mostly in isolated and uncoordinated experiences, often in the context of clinical trials. While general recommendations are probably easy to disseminate, there is still a lack of a structured and rational approach to tailored prevention, both within memory clinics and at a population level. This section will focus on a new concept of preventive services named “Brain Health Services” (BHS), emphasizing ongoing instances of such services and reporting the inception of our own experience in Monza. 

It must be acknowledged that preventive lifestyle changes are not easy to sustain even when proposed within a specialized preventive center. Therefore, systemic socio-political changes at a population level should make healthy choices easier and drive positive changes also in low-risk individuals who are not likely to come to medical attention [5]. The implementation of BHS, pursuing a “high-risk approach”, needs to be accompanied by a “population-wide approach” if we want to see meaningful changes and maintain equity [9]. Moreover, the reductions in a few relevant risk factors—such as education or air pollution [75]—greatly depend on governmental policies and cannot be addressed even by extremely specialized neurological services. For instance, the Norwegian approach to brain health—combining primordial, primary, and secondary prevention—has demonstrated the potential to reduce the incidence of dementia, stroke and ischemic heart disease (the so-called “triple threat”) [76,77,78]. Nevertheless, it does not specifically include a dedicated BHS to our knowledge. Interestingly, up to 70% of Norwegian citizens are aware of the potential of dementia risk reduction; nevertheless, some risk factors are more frequently recognized than others [79]. It is possible that the three decades of efforts put in place by the Norwegian government have raised awareness in the general population, leading to an incentive for virtuous lifestyle changes. Similar awareness has been reported in a German sample of elderly subjects with almost 40% interested in dementia risk reduction [80]. 

Advocacy by neurologists for brain health with political entities and decision-makers is crucial, encouraging them to address and prioritize these aspects and setting policy agendas [81]. It should also be noted that efforts for brain health should not be limited to resource-rich countries but rather part of global actions pursuing what has been proposed as a “brain healthy diplomacy” [54,82]. 

BHS have been recently proposed by the European Task Force for Brain Health Services panel of experts. A six-part user manual for BHS was published in 2021, including papers on dementia risk profiling, risk communication and reduction, and cognitive enhancement [83,84,85,86,87,88]. This was followed this year by a paper by Frisoni et al., tracing the roadmap of such services [89]. While we refer the reader to these excellent papers for a more comprehensive overview of BHS, the main principles underlying them are quite simple. Access to BHS is—for the time being—mostly reserved for SCD subjects, and their purpose is to provide a thorough assessment of dementia risk based on state-of-the-art evidence and rational use of biomarkers [84], communicate such risk according to established protocols [85] and apply appropriate strategy to reduce the risk. For subjects carrying a low risk of dementia, cognitive enhancement strategies may be put in place, although further evidence should be gathered on the subject [87]. For those at high risk, personalized multidomain interventions aimed at the reduction of such risk could be proposed [86]. In the future, disease-modifying therapies, such as the recently FDA-approved Lecanemab, might be proposed if studies will show clinical benefits in this population [35,83,89,90]. 

BHS substantially differ from memory clinics, and have distinct concepts. Instead of a diagnostic process, users undergo a risk assessment; instead of treatment, subjects are offered strategies to mitigate risk and possibly cognitive enhancement in the future. General practitioners may lack the expertise to provide risk estimates and tailored interventions beyond general recommendations; on the other hand, memory clinics are not currently designed for these subjects and have little to offer them beyond reassurance and, again, general recommendations. BHS will ideally fill this gap [89]. 

Further users of BHS might be people with functional cognitive disorders and the so called “worried wells”. The former are subjects usually presenting with attentional cognitive symptoms mostly in the context of anxiety, depression, or chronic pain [91]. For these subjects, the role of BHS might mostly consist in explanation of the diagnosis, possibly with the aid of Appendix A (such as those available at https://neurosymptoms.org/en/symptoms/fnd-symptoms/functional-cognitive-symptoms/) [83]. Worried wells are subjects without cognitive complaints who are concerned for future decline in cognition and willing to preserve cognitive function through prevention [83].

An important strength of BHS will be providing scientifically sound recommendations to a population that is increasingly recognizing the importance of prevention. This is important to counterbalance the promotion of pseudoscience or questionable products that feed on this need for prevention and brain health, making billions in the process [92,93]. Thus, BHS operators may prove a reliable source of information for the population interested in prevention of cognitive decline.

In order to calculate the risk of dementia, BHS operators need to comprehensively assess risk factors and protective factors, including the twelve factors listed by the Lancet Commission [11] as well as other emerging ones as detailed previously. Risk profiling should take all of these factors into account, possibly with the aid of multidomain clinical risk scores such as CAIDE, BSDI, or ANU-ADRI [94,95,96]. It is crucial that these risk scores are used in the population in which they have been validated; for instance, CAIDE may be used for people aged 39–64, while BDSI is indicated in people over 64. They also provide different predictions, the CAIDE being at 20 years, and the BDSI up to 5 years. While such risk models are certainly useful, it is important to understand their limitations and imperfect accuracy, around 70–80% [84]. Other risk calculation tools—such as ADappt, developed at Alzheimercentrum Amsterdam—rely less on clinical factors, basing fine-tuning prediction on CSF and imaging biomarkers [97]. While CSF biomarkers may not be practical in the context of BHS, blood-based biomarkers may be promising in the future [98]. If there is familial history of AD, testing for genetic risk factors such as APOE status may be useful, while in the future, polygenic risk scores may be practical [84]. However, incorporating all these factors into a single prediction model is not an easy task, and for the moment, it might be more useful to categorize BHS users into low or high risk. More research with long follow-up and comprehensive phenotyping of participants is necessary to refine risk models.

Risk communication in BHS should ideally follow established protocols developed from evidence gathered from memory clinics, clinical trials, and research registries. In this sense, it has been shown that disclosure of APOE status or amyloid-PET results to cognitively unimpaired individuals might be a sort of preventive intervention itself, as it seems to increase the propensity of high-risk subjects to adopt a healthier lifestyle. At the same time, this did not lead to increased anxiety or depression [99,100]. Communicating risk of dementia is not an easy task, and a protocol to guide this crucial aspect has been proposed, stressing the importance of tailoring effective communication on the individual’s knowledge background [89]. Importantly, the use of visual representations of risk profiles is strongly encouraged, and these can be easily prepared by BHS operators with freely available websites such as www.iconarray.com. However, it must be noticed that evidence on risk communication mostly come from other medical settings with different challenges, such as oncology, and the protocol proposed for BHS needs to be tested and eventually refined based on context-specific evidence that will be accumulated [85].

After risk profiling in the context of BHS, subjects at high risk should be proposed personalized multidomain interventions targeting several risk factors at once with sufficient intensity (as detailed in Section 3.2) [86]. To improve adherence, a few considerations need to made: firstly, smaller changes introduced gradually may help sustain long-term adoption of a healthier lifestyle. Moreover, group activities should be encouraged to take advantage of the social component. Finally, BHS’ role will also be to monitor adherence and offer support in specific domains if needed [86]. Multidomain interventions are inherently different from generic recommendations, in that they are structured and tailored to the individual’s prevention potential. It must also be acknowledged that the implementation of structured multidomain interventions within the context of BHS poses new challenges compared to clinical trials, such as the availability of human and structural resources to deliver and guide such interventions. In this context, it is crucial to emphasize the importance of support from local policy-makers and healthcare systems in allocating and funding the necessary resources for effective prevention [83].

For the time being, BHS might be limited pilot experiences in high-income, resource-rich countries. However, future developments will extend these services to the general population and low–middle-income countries, enriching their offer with blood-based biomarkers, and if clinical development allowed it, also disease-modifying therapies and non-invasive brain stimulation for cognitive enhancement [89]. Another important issue that will need to be addressed regards reimbursement. While it is possible that this aspect will be influenced by local regulations, if accumulating evidence will prove BHS to be cost-effective ways to reduce risk of dementia, it might be expected that they will be reimbursed by healthcare payers [89]. Indeed, the Scottish government has already funded demonstrator sites within a national strategy [101], while the pilot BHS we developed in Monza were approved by our institution within our memory clinic (see Section 4.1).

As of today, only a few experiences worldwide have been implemented to shift care standards in neurology from brain diseases to brain health. Even before the proposal of BHS, a similar service has been detailed within the Center for Brain Health at NorthShore Neurological Institute in Illinois, US [102]. Another recent proposal is the Scottish Model for Brain Health Services, which probably represents the first nationwide approach to prevention, substantially overlapping with the BHS outlined by the European Task Force [101]. Other pilot experiences with BHS are ongoing in different European countries, including those at Karolinska Institut in Sweden, at the Uniklinik Köln in Germany, at BarcelonaBeta in Spain, at the Amsterdam UMC in the Netherlands, at Paris La Pitié-Salpetrière in France, and at Geneva HUG in Switzerland. A map of brain health approaches in Europe is shown in Figure 2. An international conference on BHS will take place on 8th February 2024, in Geneva, Switzerland (https://centre-memoire-hug.mynelis.com/extranet/events/1, accessed on 24 December 2023), featuring presentations on all these pilot experiences. 

### 4.1. The Monza BHS

Recently, we started a BHS in Monza, as a pilot experience provisionally nested within the memory clinic of Fondazione IRCCS San Gerardo dei Tintori. Access criteria include confirmed SCD according to the established criteria [14], as well as the so-called “well worried” and functional subtle cognitive decline. Cases are notified of our service by neurologists operating in the memory clinic or general neurological outpatient clinic of our institute. In order to enrich our cohort, we started broadening recruitment strategies through the aid of local GPs. A screening strategy may be envisioned in the future, although it would probably be difficult to implement in the general population [103]. 

Each visit follows a comprehensive semi-standardized template (available in the Appendix A in English), aiming to balance the need for standardized data collection and clinical evaluation flexibility. This template aligns with the pillars proposed by Frisoni et al. in their manual. Subjects are asked about a comprehensive range of cognitive and lifestyle domains, potentially reversible causes of cognitive impairment are looked for, and concerns regarding cognition are registered and addressed. The presence of an informant is generally encouraged. The protocol includes also an estimate of the risk of developing dementia by means of CAIDE or BDSI according to age (both can be calculated with the aid of a shinyapp available through a link within the template) and risk communication. Subjects with low risk are then referred to structural brain imaging and annual standardized neuropsychological tests, while subjects at high risk are preferentially enrolled in the ongoing CAPE study (https://clinicaltrials.gov/study/NCT05756270, accessed on 28 November 2023), where they undergo brain MRI, brain FDG-PET, lumbar puncture for CSF analysis at baseline, and annual neuropsychological tests. Further exams are guided by symptoms and/or history, such as polysomnography for obstructive sleep apnea syndrome. Finally, personalized preventive strategies based on risk factors emerging from the visit are proposed, including for instance a healthy diet through referral to a nutritionist [104], physical activity based on the adaptation of relevant WHO guidelines [105,106], cognitive stimulation, management of cardiovascular risk factors [27], and sleep counseling. Subjects are also offered the opportunity to enter ongoing pharmacological and non-pharmacological preventive trials at our center, or observational studies on the early phases of neurodegenerative diseases, if they wish so. If and when subjects convert to MCI according to established clinical criteria [107], they will be offered the possibility to enter the regular care path in our memory clinic. The general pathway of care at our institution is shown in Figure 3. 

A visit at our BHS typically lasts one hour in order to spend enough time to ensure maximum understanding of the condition [103]. While this may not be feasible in busy memory clinics, it could be considered a suitable approach for specialized services such as BHS. 

Our cohort is by design not enriched for preclinical AD, but rather reflects a heterogeneous population of SCD seeking medical help and interested in preventive strategies. Indeed, subjects are referred by other neurologists of our institution and are informed about the nature and purposes of our BHS. Therefore, an inherent bias in our cohort is that it reflects a particular sub-population of SCD subjects, which may not be representative of the general SCD population. Bearing that in mind, it may be worth noticing that this sub-population may be the one for which preventive strategies are easier to implement, as the subjects themselves are, at least in theory, willing to adopt them. Moreover, it is possible that this particular population might be the only relevant one unless we find effective strategies to engage SCD subjects who do not seek medical help and/or are not willing to consider lifestyle changes. 

## 5. Conclusions

While the concept of brain health has gained recognition, there is an imperative need for intensified research in primary prevention strategies. The intricate nature of brain health, coupled with its profound impact on the healthcare system’s organization, underscores the complexity of this field. As we confront the escalating global burden of neurological and non-communicable diseases, a focused and collaborative effort is essential. Striking a balance between understanding the multifaceted determinants of brain health and implementing effective preventive measures is crucial. This collective endeavor will not only enhance individual well-being but also contribute significantly to reducing the overall burden on healthcare systems worldwide. 

## Figures and Tables

**Figure 1 brainsci-14-00055-f001:**
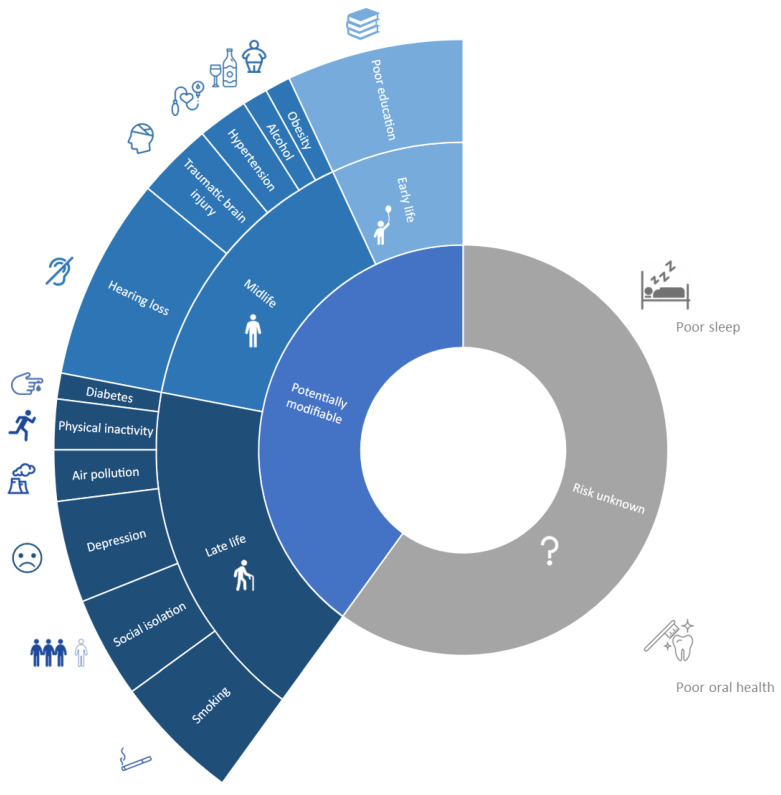
Potentially modifiable risk factors for dementia, according to 2020 Lancet Commission report. Attributable risk for poor sleep and oral health are still controversial.

**Figure 2 brainsci-14-00055-f002:**
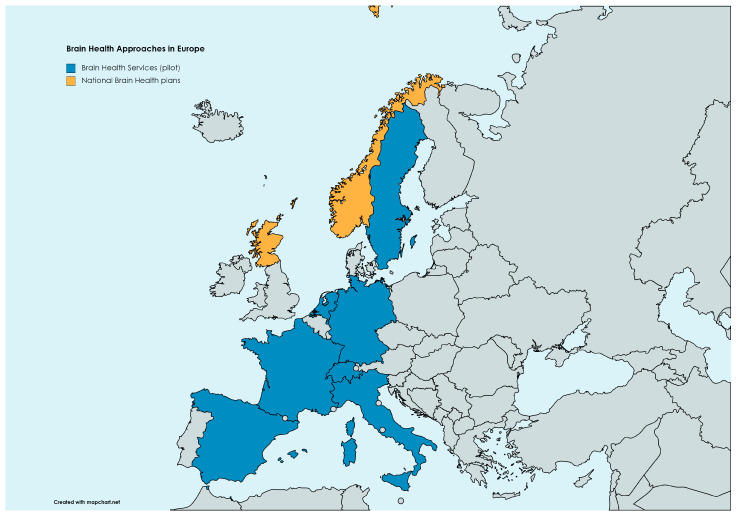
Brain health approaches in Europe as of today. Brain health services are limited to pilot experiences in Amsterdam (the Netherlands), Barcelona (Spain), Geneva (Switzerland), Köln (Germany), Monza (Italy), Paris (France), and Stockholm (Sweden). National brain health plans are ongoing in Scotland and Norway.

**Figure 3 brainsci-14-00055-f003:**
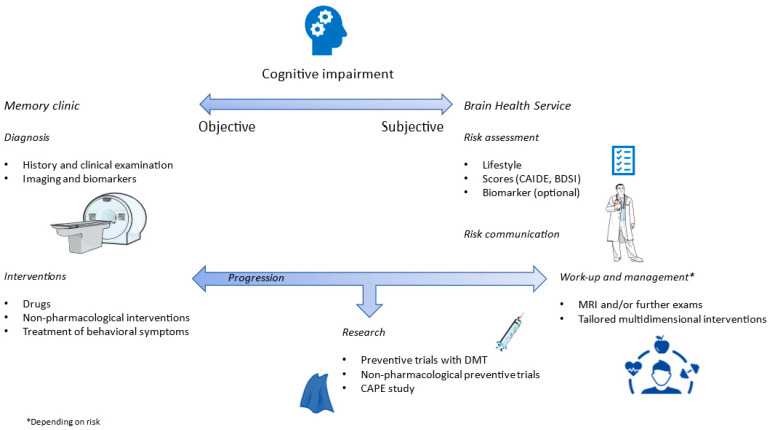
Care pathway for cognitive impairment at our institution.

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
