# Peer review of "Brain Health and Cognition in Older Adults: Roadmap and Milestones towards the Implementation of Preventive Strategies"

_brainsci, 2024, doi:10.3390/brainsci14010055_

Round 1

Reviewer 1 Report

Comments and Suggestions for Authors

This comprehensive narrative review critically explores the evolving concept of Brain Health. Emphasizing the urgency of preventive strategies, especially for older adults, the article narrows its focus on individuals with subjective cognitive decline. It provides a nuanced discussion on multidomain personalized preventive approaches, backed by empirical evidence on modifiable risk factors. The inclusion of Brain Health Services as a pivotal component in risk assessment and tailored interventions is commendable. The article also offers insights into a BHS pilot in Monza, Italy, elucidating its operational structure. Here are some concerns:

1. One notable concern pertains to the breadth of the article's title. The discourse, while addressing critical aspects of the overarching theme, tends to be somewhat succinct, potentially lacking the desired depth commensurate with the expansive title.

2. The perspective posited in the article, centering on individuals with SCD as the focal point for preventive strategies, is not entirely aligned with prevailing perspectives. SCD subjects often exhibit a degree of cerebral compromise, with such deficits often manifesting 5-10 years prior to cognitive impairment. Implementing preventive strategies for cognitively normal middle-aged and older adults for preventive interventions also holds great promise for brain health.

Ref:

Chang YK, Erickson KI, Aghjayan SL, Chen FT, Li RH, Shih JR, Chang SH, Huang CM, Chu CH. The multi-domain exercise intervention for memory and brain function in late middle-aged and older adults at risk for Alzheimer's disease: A protocol for Western-Eastern Brain Fitness Integration Training trial. Front Aging Neurosci. 2022 Aug 18;14:929789.

Halloway S, Wilbur J, Schoeny ME, Arfanakis K. Effects of Endurance-Focused Physical Activity Interventions on Brain Health: A Systematic Review. Biol Res Nurs. 2017 Jan;19(1):53-64.

Jin Y, Lin L, Xiong M, Sun S, Wu SC. Moderating effects of cognitive reserve on the relationship between brain structure and cognitive abilities in middle-aged and older adults. Neurobiol Aging. 2023 Aug;128:49-64.

3. 'While cognitive reserve is built through education and activities in the first part of life'

While cognitive reserve traditionally emphasizes the role of education and cognitive activities in early life, it is essential to note that cognitive reserve is not solely built during the initial phases of life. Contemporary perspectives highlight its ongoing relationship with cognitive stimulation and activities throughout the lifespan, extending into middle and late adulthood.

Ref:

Mungas D, Early DR, Glymour MM, Zeki Al Hazzouri A, Haan MN. Education, bilingualism, and cognitive trajectories: Sacramento Area Latino Aging Study (SALSA). Neuropsychology. 2018 Jan;32(1):77-88.

Hall CB, Lipton RB, Sliwinski M, Katz MJ, Derby CA, Verghese J. Cognitive activities delay onset of memory decline in persons who develop dementia. Neurology. 2009 Aug 4;73(5):356-61.

4. The author's presentation of BHS is somewhat cursory. It is advisable to align this exposition with earlier discussions on research limitations, emphasizing distinctions between BHS and preceding studies. A more intricate exploration of innovative design elements would accentuate BHS's unique features and contributions in comparison to prior research.

5. Having thoroughly reviewed the entire manuscript, it appears that the focal point lies in the Monza BHS. However, the article title, abstract, and overall organization, potentially hindering readers from effectively grasping the central theme. It is recommended that the author consider a structural adjustment to align these elements coherently, ensuring that the prominence of the Monza BHS is reflected accurately throughout the paper.

Author Response

This comprehensive narrative review critically explores the evolving concept of Brain Health. Emphasizing the urgency of preventive strategies, especially for older adults, the article narrows its focus on individuals with subjective cognitive decline. It provides a nuanced discussion on multidomain personalized preventive approaches, backed by empirical evidence on modifiable risk factors. The inclusion of Brain Health Services as a pivotal component in risk assessment and tailored interventions is commendable. The article also offers insights into a BHS pilot in Monza, Italy, elucidating its operational structure. Here are some concerns:

  1. One notable concern pertains to the breadth of the article's title. The discourse, while addressing critical aspects of the overarching theme, tends to be somewhat succinct, potentially lacking the desired depth commensurate with the expansive title.

We thank the reviewer for taking time in evaluating our manuscript. We acknowledge that the original manuscript lacked an in-depth analysis of the points kindly highlighted by the reviewer. As we modified and extensively integrated the paper, we hope to have matched the title expectations.

  1. The perspective posited in the article, centering on individuals with SCD as the focal point for preventive strategies, is not entirely aligned with prevailing perspectives. SCD subjects often exhibit a degree of cerebral compromise, with such deficits often manifesting 5-10 years prior to cognitive impairment. Implementing preventive strategies for cognitively normal middle-aged and older adults for preventive interventions also holds great promise for brain health.

Ref:

Chang YK, Erickson KI, Aghjayan SL, Chen FT, Li RH, Shih JR, Chang SH, Huang CM, Chu CH. The multi-domain exercise intervention for memory and brain function in late middle-aged and older adults at risk for Alzheimer's disease: A protocol for Western-Eastern Brain Fitness Integration Training trial. Front Aging Neurosci. 2022 Aug 18;14:929789.

Halloway S, Wilbur J, Schoeny ME, Arfanakis K. Effects of Endurance-Focused Physical Activity Interventions on Brain Health: A Systematic Review. Biol Res Nurs. 2017 Jan;19(1):53-64.

Jin Y, Lin L, Xiong M, Sun S, Wu SC. Moderating effects of cognitive reserve on the relationship between brain structure and cognitive abilities in middle-aged and older adults. Neurobiol Aging. 2023 Aug;128:49-64.

We thank the reviewer for this excellent suggestion. We inserted a new paragraph (3.1.2) on cognitively normal middle aged and older adults, and integrated the suggested references, as well as a brief discussion on biological effects of multi-domain interventions, as requested by reviewer 2.

  1. 'While cognitive reserve is built through education and activities in the first part of life'

While cognitive reserve traditionally emphasizes the role of education and cognitive activities in early life, it is essential to note that cognitive reserve is not solely built during the initial phases of life. Contemporary perspectives highlight its ongoing relationship with cognitive stimulation and activities throughout the lifespan, extending into middle and late adulthood.

We toned down our statement, rephrasing as “While cognitive reserve may be primarily built through education and activities in the first part of life, brain resilience may be increased by cognitive training and socialization even in late adulthood”. We then emphasized the role of cognitive activities in the following sentences, including the suggested references.

Ref:

Mungas D, Early DR, Glymour MM, Zeki Al Hazzouri A, Haan MN. Education, bilingualism, and cognitive trajectories: Sacramento Area Latino Aging Study (SALSA). Neuropsychology. 2018 Jan;32(1):77-88.

Hall CB, Lipton RB, Sliwinski M, Katz MJ, Derby CA, Verghese J. Cognitive activities delay onset of memory decline in persons who develop dementia. Neurology. 2009 Aug 4;73(5):356-61.

  1. The author's presentation of BHS is somewhat cursory. It is advisable to align this exposition with earlier discussions on research limitations, emphasizing distinctions between BHS and preceding studies. A more intricate exploration of innovative design elements would accentuate BHS's unique features and contributions in comparison to prior research.

We agree with the reviewer that the BHS presentation is a bit scarce and superficial. We extensively enriched this section, including a detailed explanation of BHS features and their differences from memory clinics, as well as the challenges they stand to face and current limitations.

  1. Having thoroughly reviewed the entire manuscript, it appears that the focal point lies in the Monza BHS. However, the article title, abstract, and overall organization, potentially hindering readers from effectively grasping the central theme. It is recommended that the author consider a structural adjustment to align these elements coherently, ensuring that the prominence of the Monza BHS is reflected accurately throughout the paper.

It was not our intention to give such a central role to the Monza BHS, which should have ideally been only an example of a pilot experience, albeit detailed. However, we realized that the first version of our paper lacked enough breadth to counterbalance the description of our service. With the modifications we introduced and the relative expansion of the other sections, we think we have now aligned the presentation of all the covered topics to the relative importance we intended to give them.

Reviewer 2 Report

Comments and Suggestions for Authors

In the manuscript the authors have addressed the evolving concept of brain health.  The need for preventive strategies for cognitive decline and population that might benefit from such approaches.  The authors have also highlighted the BHS and dealt with their proposed role to reduce risk of dementia.  Further, the authors emphasize the need for collaborative effort at a global scale to address the determinants for brain health.

The manuscript has collected some data from studies and does not deal extensively with the determinants of brain health.

The following sections I feel will be good addition to the manuscript.

The role of cellular metabolism at old age and insights from animal studies

The role of nutrition and how it modulates healthy brain.

Animal and preclinical studies will be highly useful.

A discussion on integrating the data of reduced availability of nutrients on brain health will be useful.

Comments on the Quality of English Language

The english is fine.

Author Response

In the manuscript the authors have addressed the evolving concept of brain health.  The need for preventive strategies for cognitive decline and population that might benefit from such approaches.  The authors have also highlighted the BHS and dealt with their proposed role to reduce risk of dementia.  Further, the authors emphasize the need for collaborative effort at a global scale to address the determinants for brain health.

The manuscript has collected some data from studies and does not deal extensively with the determinants of brain health. The following sections I feel will be good addition to the manuscript.

The role of cellular metabolism at old age and insights from animal studies

The role of nutrition and how it modulates healthy brain.

Animal and preclinical studies will be highly useful.

A discussion on integrating the data of reduced availability of nutrients on brain health will be useful.

We thank the reviewer for their insights and time in reviewing our manuscript. We have added a brief discussion on the preclinical data concerning the different determinants of brain health in the new 3.3 section. However, we feel that shifting the discourse on nutrition and cellular metabolism may fall beyond the scope of our paper, which is more reflective of the current endeavors in the implementation of preventive strategies, with an important clinical perspective.

Reviewer 3 Report

Comments and Suggestions for Authors

The current manuscript was a narrative review and focused on the topic of Brain Health. In particular, the cognitive aspects of brain health decline with age. The manuscript covered prevention of cognitive decline and various risk factors. In addition,  Brain Health Services (BHS) and their role in risk assessment, risk communication, and tailored interventions to reduce the risk of dementia was covered. Finally the authors covered their own BHS in Italy and discussed its operational structure and care pathways.

 Overall, this is a novel narrative review and I don’t think one like it has been published before. It was well-written overall and easy to understand. I think the study adds to the literature on the topics involved. The narrative review will be of interest to readers of the journal and researchers is several related fields.

 I don’t think the narrative had any fatal flaws or major weaknesses. Thus, I only have some minor comments and suggestions for changes.

 There are some run on sentences in places where several large clauses or sentences are combined into a single sentence with multiple comments. Please reword. For instance, the abstract has at least 2 of these.

  1.  
  2. There needs to be a space before the bracket used for each citation in the text. Right now there is no space before the bracket for any of them. Thus, the word and the brackets/citation are merged and need a space.
  3. The words in Figure 1 would benefit from a larger font size if possible.
  4. Lines 242 and 243. Should it be Geneva?
  5. The bibliography has errors. Sometimes the first letter of each word in the title of the papers is capitalized and sometimes it is not. For instance compare Refs 2 and 3. There are many other examples. Please proofread the bibliography.
Comments on the Quality of English Language

minor rewording of some sentences, some run on sentences

Author Response

The current manuscript was a narrative review and focused on the topic of Brain Health. In particular, the cognitive aspects of brain health decline with age. The manuscript covered prevention of cognitive decline and various risk factors. In addition,  Brain Health Services (BHS) and their role in risk assessment, risk communication, and tailored interventions to reduce the risk of dementia was covered. Finally the authors covered their own BHS in Italy and discussed its operational structure and care pathways.

 Overall, this is a novel narrative review and I don’t think one like it has been published before. It was well-written overall and easy to understand. I think the study adds to the literature on the topics involved. The narrative review will be of interest to readers of the journal and researchers is several related fields.

 I don’t think the narrative had any fatal flaws or major weaknesses. Thus, I only have some minor comments and suggestions for changes.

 There are some run on sentences in places where several large clauses or sentences are combined into a single sentence with multiple comments. Please reword. For instance, the abstract has at least 2 of these.

We thank the reviewer for taking time in evaluating our paper. We followed the suggestion of the reviewer and reworded lengthy passages to improve readability and clarity.

  1.  
  2. There needs to be a space before the bracket used for each citation in the text. Right now there is no space before the bracket for any of them. Thus, the word and the brackets/citation are merged and need a space.

We added the needed space before the bracket.

  1. The words in Figure 1 would benefit from a larger font size if possible.

As the figure reflects the relative contribution of each factor graphically, increasing the font size results in a displacement of the text beyond the border of each section (especially for “hypertension” or “physical inactivity”). The font size we selected was the largest possible to avoid this effect. However, we enlarged the whole figure to improve readability.

  1. Lines 242 and 243. Should it be Geneva?

We changed the name of the city using the English rather than the French spelling.

  1. The bibliography has errors. Sometimes the first letter of each word in the title of the papers is capitalized and sometimes it is not. For instance compare Refs 2 and 3. There are many other examples. Please proofread the bibliography.

This depended on the specific “capitalization” policy of each journal in our citation manager software. We proofread the whole bibliography and consistently removed capitalized letters.

Round 2

Reviewer 1 Report

Comments and Suggestions for Authors

We greatly appreciate the author's diligent efforts in addressing the previously identified issues; however, I still harbor one minor reservations regarding certain aspects of the manuscript.

1.Not all middle-aged and elderly individuals will undergo the SCD phase before experiencing dementia. The focus of the article revolves more around interventions during the SCD stage. Would it be more appropriate to change the title to 'Brain Health and Cognition in SCD: Roadmap and Milestones towards the Implementation of Preventive Strategies'?

Author Response

We agree with the reviewer on the focus of the article, and changed the title accordingly, as suggested. Thank you for reviewing our manuscript. 

Reviewer 2 Report

Comments and Suggestions for Authors

The Authors have addressed the comments.  No more comments

Author Response

Thank you very much for reviewing our manuscript.